# EEG Characterization of the Alzheimer’s Disease Continuum by Means of Multiscale Entropies

**DOI:** 10.3390/e21060544

**Published:** 2019-05-28

**Authors:** Aarón Maturana-Candelas, Carlos Gómez, Jesús Poza, Nadia Pinto, Roberto Hornero

**Affiliations:** 1Biomedical Engineering Group, E.T.S.I. de Telecomunicación, Universidad de Valladolid, 47011 Valladolid, Spain; 2Instituto de Investigación en Matemáticas (IMUVA), Universidad de Valladolid, 47011 Valladolid, Spain; 3Instituto de Neurociencias de Castilla y León (INCYL), Universidad de Salamanca, 37007 Salamanca, Spain; 4Instituto de Patologia e Imunologia Molecular da Universidade do Porto (IPATIMUP), 4200-135 Porto, Portugal; 5Instituto de Investigação e Inovação em Saúde (i3S), 4200-135 Porto, Portugal; 6Center of Mathematics of the University of Porto (CMUP), 4169-007 Porto, Portugal

**Keywords:** Electroencephalography (EEG), multiscale sample entropy (MSE), refined multiscale spectral entropy (rMSSE), Alzheimer’s disease (AD), mild cognitive impairment (MCI), AD continuum

## Abstract

Alzheimer’s disease (AD) is a neurodegenerative disorder with high prevalence, known for its highly disabling symptoms. The aim of this study was to characterize the alterations in the irregularity and the complexity of the brain activity along the AD continuum. Both irregularity and complexity can be studied applying entropy-based measures throughout multiple temporal scales. In this regard, multiscale sample entropy (MSE) and refined multiscale spectral entropy (rMSSE) were calculated from electroencephalographic (EEG) data. Five minutes of resting-state EEG activity were recorded from 51 healthy controls, 51 mild cognitive impaired (MCI) subjects, 51 mild AD patients (AD_MIL_), 50 moderate AD patients (AD_MOD_), and 50 severe AD patients (AD_SEV_). Our results show statistically significant differences (*p*-values < 0.05, FDR-corrected Kruskal–Wallis test) between the five groups at each temporal scale. Additionally, average slope values and areas under MSE and rMSSE curves revealed significant changes in complexity mainly for controls vs. MCI, MCI vs. AD_MIL_ and AD_MOD_ vs. AD_SEV_ comparisons (*p*-values < 0.05, FDR-corrected Mann–Whitney *U*-test). These findings indicate that MSE and rMSSE reflect the neuronal disturbances associated with the development of dementia, and may contribute to the development of new tools to track the AD progression.

## 1. Introduction

Alzheimer’s Disease (AD) is a progressive neurodegenerative disorder, which is associated with cognitive impairment, behavior disorders and memory loss, among others. AD is the most common cause of dementia, with an estimated 60–80% of cases [1]. In 2018, 50 million people worldwide were coexisting with some type of dementia [2]. This number is expected to be more than tripled by 2050, rising to 152 million [2]. AD can be considered a highly disruptive disorder, not only for patients, but also for families and caregivers, being the main cause of dependence and disability for older people [3]. AD is particularly costly for the society, reaching a global economic burden of about US$ 1 trillion per year in 2018 [2].

A transitional period is likely to occur between normal ageing and the diagnosis of early dementia due to AD. This state is described as mild cognitive impairment (MCI). MCI patients are usually diagnosed with slight cognitive deficits, but insufficient to categorize them as dementia patients [4]. AD may eventually develop from this point through three main stages of progressive severity. Firstly, mild AD patients (AD_MIL_) are characterized by clear-cut impairments on careful clinical interview and deficits manifestations in areas such as reduced knowledge of recent events, difficulty to recall personal information, and decreased ability to handle finances or travel. Secondly, moderate AD patients (AD_MOD_) are unable to recall important events in their lives and suffer disorientation regarding with current place and time. Finally, severe AD patients (AD_SEV_) are completely dependent for survival. Patients in this stage are generally oblivious about their surroundings, time, place and recent events. They tend to remember some but sketchy details about their long-term life experiences [5].

Nowadays, the methods for AD diagnosis are complex and mainly subjective. Some of the procedures applied include examination of medical history, neurological exams and diagnostic tests, along with other techniques [1,6]. Unfortunately, the gold standard for AD diagnosis according to the National Institute on Aging and Alzheimer’s Association (NIA-AA) criteria is the histological examination of brain tissue through biopsy or autopsy [7]. This practice confirms the presence of neurofibrillary tangles and Aβ peptide, which are strongly associated with AD [8]. For these reasons, the development of new tools for AD diagnosis appears to be relevant.

Electroencephalography (EEG) is a non-invasive technique that allows the measurement of the electrical brain activity. EEG devices can record the transient and rapid nature of brain signaling due to its high temporal resolution. In the last decades, EEG analyses with spectral and non-linear measures have provided new insights into the understanding of physiological dynamics, including alterations due to dementia. Among them, entropy measures have received great attention. Entropy estimates the randomness of a series, which is directly related with its predictability [9]. Several entropy methods have been developed, such as spectral entropy (SpecEn) [10], approximate entropy (ApEn) [11], sample entropy (SampEn) [12], fuzzy entropy (FuzEn) [13] or permutation entropy (PermEn) [14]. Remarkably, many studies have revealed a strong correlation between cognitive decline and loss of EEG irregularity [15,16,17,18,19].

Traditional entropy measures analyze a single temporal scale and, sometimes, they are not able to express the physiological dynamics of more complex processes of the brain. To address this issue, Costa et al. [20] suggested the application of multiscale entropy methods. Difference between regularity and complexity was also remarked, pointing the importance of “meaningful structural richness” [20]. From the physiological point of view, biosignal complexity is associated with the adaptation capability of a living system in a changing environment [20]. For this reason, loss of complexity has been described as a common feature of neuropathology [21,22]. The analysis of this feature requires integrative multiscale functionality, which is unable to be studied by traditional entropy methods [20]. In this regard, multiscale entropy (MSE) describes the complexity of underlying neural systems working in a wide-range of temporal scales. Costa et al. [20] proposed MSE as an extension of SampEn, basing its calculation on coarse-grained time series. Other authors have previously used MSE to examine spatiotemporal patterns of the EEG in order to assess neural background activity in AD [23,24].

Spectral analyses have also been carried out to study the AD progression. EEG signals from AD patients show a shift in the power spectral density (PSD) towards lower frequencies (theta and delta bands) and a decrease of power in alpha and beta bands [25,26]. Abásolo et al. [17] suggested that spectral analyses might be complementary with non-linear analyses in AD categorization. For this reason, the combination of both kinds of measures may lead to obtain a better description of its disturbances on the dynamics of neural activity. Entropies based on extensive information, such as Shannon and Rényi entropies, and based on non-extensive information, as Tsallis entropy, have been used to quantify the irregularity of the spectrum in AD [27]. Other studies have revealed lower SpecEn values in AD patients against controls [17,18]. Even though plenty of work has been conducted in this regard, the application of SpecEn from a multitemporal-scale approach is very limited. Humeau-Heurtier et al. [28] proposed the use of multiscale SpecEn (MSSE) to microvascular data to distinguish young from elderly subjects. MSSE measures the irregularity of the PSD of the coarse-grained versions of a signal, calculated on different scale factors. This study also provides the algorithm to compute the refined version of MSSE (rMSSE), improving its robustness against aliasing. The application of this measure may reveal complexity alterations of EEG signals from a spectral analysis standpoint. To the best of our knowledge, rMSSE has not been previously applied to characterize the brain activity in AD.

The aim of this study was to ascertain the relation between EEG signal complexity and the degree of cognitive impairment among five subject groups of different AD severity. For this purpose, multitemporal-scaled entropy measures in time and frequency domains were calculated. We also proposed to assess complexity in a quantitative way, making use of global analysis parameters. Additionally, we tested the classification power of the combination of MSE and rMSSE in order to distinguish subjects of each group and, hence, assessing the diagnostic ability of the procedure. The results obtained in this research could offer new insights into how the structural complexity of the EEG may be affected by neurodegeneration caused by dementia.

## 2. Materials

### 2.1. Subjects

Two hundred fifty-three subjects took part in this study: 51 control subjects, 51 MCI subjects, 51 AD_MIL_ patients, 50 AD_MOD_ patients and 50 AD_SEV_ patients. Every subject with MCI or dementia due to AD was diagnosed and categorized according to the criteria of the NIA-AA [7]. Minimental State Examination (MMSE) scoring was used to evaluate the cognitive mental state of the patients [29].

For AD and MCI patients, inclusion criteria included ages older than 65 and a diagnosis of dementia or MCI due to AD, respectively. Exclusion criteria included the presence of atypical signs in evolution, history of active or treatment neoplasia, history of recent surgery or hypercatabolic states, chronic alcoholism or any indications of components of vascular pathology in clinical history. On the other hand, control subjects were required to meet the following criteria to participate in the study. Inclusion criteria were ages older than 65 and MMSE scores higher than 27. Exclusion criteria included the presence of neurological history or major psychiatric disorders. All subjects and caregivers gave a written consent to participate in this study, according to the recommendations of the Code of Ethics of the World Medical Association (Declaration of Helsinki). The protocol was approved by The Ethics Committee at the Porto University (Porto, Portugal). Table 1 provides a summary of the demographic data of the participants.

### 2.2. EEG Recording

For each subject, five minutes of EEG activity were acquired with a 19-channel Nihon Kohden Neurofax JE-921A EEG System at electrodes F3, F4, F7, F8, Fp1, Fp2, T3, T4, T5, T6, C3, C4, P3, P4, O1, O2, Fz, Cz, and Pz of the international 10–20 system, with common average reference. Sampling frequency was established at 500 Hz. Subjects were asked to stay in a relaxed state with eyes closed in a noise-free environment to minimize artifact presence. Researchers made sure to avoid drowsiness of the participants during the procedure.

EEG data were stored in a personal computer as ASCII files and were preprocessed according to the following steps [30,31]: (i) mean removal; (ii) 50 Hz notch filtering; (iii) bandpass filtering with a Hamming window between 0.4 and 98 Hz; (iv) independent component analysis (ICA), to remove components associated with myographic, cardiographic and oculographic noise; (v) segmentation into 5 s epochs; and (vi) visual rejection of epochs contaminated by artifacts. An average number of 38.94 ± 13.12 (mean ± SD) artifact-free epochs per subject were selected. Every digital process in this study was carried out with MATLAB^®^ (R2017a version, Mathworks, Natick, MA).

## 3. Methods

### 3.1. Multiscale Sample Entropy

MSE is a non-linear technique that allows perceiving the irregularity of a time series at different temporal scales to assess complexity. Costa et al. [21] pointed out that traditional methods (i.e., single-scale entropy measures) fail to quantify complexity in physiologic dynamics. To overcome this drawback, MSE was proposed, based on the calculation of SampEn at different time scales. MSE algorithm is computed through two steps: a coarse-grained procedure and the entropy calculation [20,21]. The algorithm is computed as follows:Firstly, given a one-dimensional discrete time series, [$x_{1},\ldots ,x_{i},\ldots ,x_{N}$], successive coarse-grained time series $Y^{\tau }$ are built according to the scale factor τ. This process is performed by calculating the average value of the data points from non-overlapping windows of length τ [20,24]:
(1)$$Y_{j}^{\tau }=\frac{1}{\tau }\sum_{i=(j-1)\tau +1}^{j\tau }x_{i},\;1\leq j\leq \frac{N}{\tau }=N^{\left(\tau \right)}.$$It is worth noting that the number of samples for each subsequent coarse-grained series is reduced by a factor of τ, hence $Y^{1}$ resulting in a sequence equal to the original time series.Afterwards, SampEn is calculated for each $Y_{j}^{\tau }$. SampEn algorithm is detailed in the contribution of Richman and Moorman [12]. SampEn is a non-linear measure that allows assessing the degree of irregularity of a signal [12], developed as an improvement of ApEn [11,12]. SampEn accounts as an embedding entropy, the calculation of which is based on the similarity with a delayed version of the time series itself [32]. Given *m*, a positive integer, and *r*, a positive real number, SampEn is defined as the negative logarithm of the conditional probability that two sequences of length *m* from the time series are similar within a *r* threshold at the next point, excluding self matches [12]. Setting tolerance as *r* times the standard deviation of the original time series grants robustness to variations, such as magnification, reduction or shift by a constant [20,24]. Setting the values of *r* and *m* is crucial in the performance of SampEn. However, there are no absolute guidelines to optimize these variables [24]. It has been found that excessively low *r* values may cause the calculation to fail, while excessively high values may introduce some bias [24,33]. Previous SampEn studies obtained good statistical reproducibility setting *m* = 1 and *r* = 0.25 for sequences larger than 100 points [24,33]. For this reason, the maximum scale factor was established to $\tau_{MAX}$ = 25, implying a minimum length of 100 samples for any coarse-grained time series.

This measure allows estimating EEG irregularity in the time domain (SampEn for scale 1, but also for other scale factors) as well as complexity. Different approaches have been proposed for the complexity estimation from MSE profiles, such as slope values [24] and areas under the curve [34,35]. In this study, both average slope values and areas under MSE curves were used to estimate EEG complexity. These parameters were calculated for the whole scales range as well as for low and high scale factors. In MSE, low scales were considered scale factors from 1 to 10, and high scales were considered scale factors from 11 to 25. This division was established because control group tend to decrease at this particular scale.

### 3.2. Refined Multiscale Spectral Entropy

In this study, we applied rMSSE to complement the information provided by MSE calculations. Humeau-Heurtier et al. [28] proposed to evaluate cardiovascular data on different time scales by means of rMSSE. This refined version of the MSSE method enhances the robustness against aliasing and, hence, spurious oscillations between 0 Hz and the filter cutoff frequency due to downsampling [36]. rMSSE is built from the multiscale application of the SpecEn. SpecEn is a measurement of irregularity of a signal according to its PSD. Powell and Percival defined the term, which provides a measure of the distribution of the frequency components [10]. Analogously with MSE, SpecEn was applied to each coarse-grained series previously calculated. The rMSSE algorithm is computed as follows [28]:Given a one-dimensional discrete time series, [$x_{1},\ldots ,x_{i},\ldots ,x_{N}$], a coarse-grained process is applied using the method described in Equation (Equation 1).The PSD of the coarse-grained time series $Y^{\tau }$ is calculated. The PSD consists of the distribution of power into frequency components. In this work, the PSD was obtained by calculating the fast Fourier transform (FFT) of the autocorrelation function of the EEG data. For each scale factor τ, the normalized PSD (nPSD) of the coarse-grained time series is computed as $nPSD_{\tau }\left(f_{j}\right)$, where $1\leq j\leq \frac{N}{2\tau }$, being $\frac{N}{2\tau }$ the length of the power spectrum of $Y^{\tau }$. The value of $\tau_{MAX}$ for the rMSSE study was established to 25 to maintain consistency with the MSE analysis. The nPSD is evaluated as follows:
(2)$$nPSD_{\tau }\left(f_{j}\right)=\frac{PSD_{\tau }\left(f_{j}\right)}{\sum_{j=1}^{N/(2\tau )}PSD_{\tau }\left(f_{j}\right)}.$$Then, the normalized rMSSE is calculated as the Shannon entropy of the PSD:
(3)$$rMSSE_{\tau }=-\frac{1}{ln(\frac{N}{2\tau })}\sum_{j=1}^{N/(2\tau )}nPSD_{\tau }\left(f_{j}\right)ln\left[nPSD_{\tau }\left(f_{j}\right)\right].$$The resulting rMSSE values are within the range between 0 and 1, assuming the lowest value according to a signal with a unique frequency component (i.e., a sinusoidal signal), and the highest value corresponding to a signal with an equally distributed power in all frequencies (i.e., white noise).

Analogously with MSE analysis, rMSSE slope values and areas under the rMSSE curves were calculated to assess complexity attending to EEG frequency alterations. In this case, we considered scale factors from 1 to 3 as low scales, due to the particular shape that all groups show for the aforementioned scales, while scale factors from 4 to 25 were considered as high scales.

### 3.3. Statistical and Classification Analyses

To study normality and homoscedasticity of the data, Kolmogorov-Smirnov and Shapiro–Wilk tests were applied. As our results did not meet parametric assumptions, the evaluation of statistical differences among the five groups was performed by Kruskal–Wallis test, while the evaluation of statistical differences between pairs of groups along AD continuum was performed by Mann–Whitney *U*-test. A false discovery rate (FDR) correction was applied to comparisons between multiple classes to avoid falsely rejected hypotheses [37].

Additionally, a classification procedure was performed to study the discriminant capability of MSE and rMSSE. The 62 classification features were the averaged MSE and rMSSE values at each scale factor, averaged slope values of MSE and rMSSE for all, low and high scales, and areas under MSE and rMSSE curves for all, low and high scale factors. Firstly, a stepwise multilinear regression (SMR) with a conditional forward selection approach was employed to select the most optimal features for classification. The feature selection process was carried out making use of a leave-one-out (LOO) cross validation (CV) scheme to avoid bias in the results [38]. Afterwards, a quadratic discriminant analysis (QDA) was used. QDA has advantages over linear discriminant analysis (LDA). While LDA assumes data homocedasticity and normality to model each class-conditional density function for an input feature, QDA does not [39]. QDA classifies each class obeying quadratic decision boundaries between classes, while LDA discriminates by a linear decision threshold, thus minimizing misclassification [39].

## 4. Results

### 4.1. Irregularity Analyses

MSE and rMSSE were calculated for 51 control subjects, 51 MCI subjects, 51 AD_MIL_ patients, 50 AD_MOD_ patients, and 50 AD_SEV_ patients. Figure 1 and Figure 2 illustrate the grand-average of MSE and rMSSE values along all temporal scales for each group. FDR-corrected Kruskal–Wallis test was calculated to ascertain statistical differentiation among groups, obtaining *p*-values lower than 0.05 for all temporal scales with both measures. MSE figure exhibited an increasing tendency for each group at low temporal scales, while rMSSE adopted the same trend for scale factors τ≥ 3. Groups associated with healthier cognitive states tend to stabilize or even decrease at higher temporal scales. For scales 4–8 for MSE, and scales 2–7 for rMSSE, groups appear sorted by AD stages.

FDR-corrected Mann–Whitney *U*-tests were computed between all consecutive stages of AD to categorize the disease progression. The results are indicated on the top of Figure 1 and Figure 2 . Several observations from the statistical analyses were noticed. Firstly, MSE revealed significant differences between controls and MCI (*p*-values < 0.05, FDR-corrected Mann–Whitney *U*-test) for scale factors 1, 2, 3 and from 13 to 25, while rMSSE showed statistically significant differences for scale factors 19–25. Secondly, only MSE can detect the alterations for MCI vs. AD_MIL_ comparison (scale factors 6–17). Thirdly, neither MSE nor rMSSE reported significant differences for AD_MIL_ vs. AD_MOD_ comparison. Finally, AD_MOD_ and AD_SEV_ showed differentiation for scale factors from 5 to 22 for MSE and higher than 2 for rMSSE.

### 4.2. Complexity Analyses

To quantify EEG complexity, averaged slope values and areas under the MSE and rMSSE curves were calculated. Figure 3 and Figure 4 display the distribution of these parameters for all the scale factors and also for low and high scales. Complexity parameters derived from MSE profiles (Figure 3), revealed statistically significant differences mainly for controls vs. MCI, MCI vs. AD_MIL_ and AD_MOD_ vs. AD_SEV_ comparisons, but also for AD_MIL_ vs. AD_MOD_ when the slope values for high scales were analyzed. On the other hand, areas under rMSSE curves detected significant differences between AD patients at moderate stage and patients at severe stage (see Figure 4a–c). Finally, statistically significant differences were reported between controls and MCI subjects for the slope values of rMSSE curves (see Figure 4d–f).

### 4.3. Classification Analysis

A classification analysis was carried out in order to assess whether MSE and rMSSE values could discriminate between groups at different stages of AD. Throughout this process, 253 SMR LOO-CV iterations were performed, excluding a different subject from the training set in each fold. Afterwards, the subject left out was classified with the predictor built with the data from the other subjects [40]. Subsequently, the remaining subjects were classified according to the features previously selected. Feature selection results exhibit five features that were predominantly selected throughout the CV process: scale factors 12 and 24 of MSE, averaged MSE slopes at high temporal scales, scale factor 2 of rMSSE, and averaged rMSSE slopes at high temporal scales. The classifier employed in this study was QDA. Table 2 shows the confusion matrix corresponding to the classification results (Cohen–Kappa, CK = 0.254). Additional classification computations were carried out with MSE features and rMSSE features separately; however, the combination of both approaches outperformed either of them individually (CK = 0.219 for MSE, CK = 0.200 for rMSSE).

The AD groups were classified against subjects which did not suffer from AD with an accuracy of 69.7% (sensibility = 0.818, specificity = 0.585). In addition, classification of the control group vs. AD group, excluding MCI subjects, was performed obtaining an accuracy of 79.1% (sensibility = 0.888, specificity = 0.523). It is worth mentioning that, if the classifier predicts a subject as an AD_MIL_, AD_MOD_ or an AD_SEV_ patient, the subject suffers, at least, from MCI with roughly a 90% chance.

## 5. Discussion

EEG background activity was analyzed from 51 control subjects, 51 MCI subjects, 51 AD_MIL_ patients, 50 AD_MOD_ patients, and 50 AD_SEV_ patients by means of MSE and rMSSE. Our main goal was to characterize the progression of AD through irregularity and complexity assessment. For this purpose, multiscale entropy analyses were applied to the EEG data.

### 5.1. Multiscale Entropies for Irregularity Estimation

MSE and rMSSE results at scale 1 (i.e., conventional definitions of SampEn and SpecEn) show a decrease of irregularity in AD patients comparing with control subjects as it can be observed in Figure 1 and Figure 2. Previously, entropy measures have been used in EEG irregularity evaluation, being the lowest entropy values associated to patients with dementia [17,18,41]. The decreased irregularity in AD patients might be due to widespread neuronal death, alongside with deficit in synapses efficiency due to both neurotransmitter synthesis alterations and lack of connectivity of the local neural networks [19,42,43]. Inactivation of brain tissue due to these effects may be linked to cognitive decline, however, the exact physiological reasons are not clearly known [19].

Entropy values at other temporal scales were also assessed, obtaining remarkable similarities between MSE and rMSSE. For instance, both measures showed lack of statistical difference of EEG average entropy between AD_MIL_ and AD_MOD_ subjects. Furthermore, AD_MOD_ vs. AD_SEV_ groups comparison reported the highest discrimination among all temporal scales. Remarkably, control and MCI groups exhibited statistical differences in both measures at the highest temporal scales. This distinction is particularly interesting, since it may be useful to detect early symptoms of the AD and, thus, allow early and more effective therapies. Our results show a weak separation between study groups at scale 1, widening at larger temporal scales. This observation agrees with other research on the field [44,45,46]. MSE and rMSSE curves also revealed that the five study groups appeared sorted by the degree of severity for some scale factors. This supports the idea of certain temporal scales being better than others for distinguishing groups, which can only be studied by multiscale methods.

### 5.2. Multiscale Entropies for Complexity Estimation

Two measures derived from MSE and rMSSE profiles may be used to estimate the EEG complexity: averaged slope values and areas under MSE and rMSSE curves. In previous research, slope values [24] and areas under MSE curves were used to elucidate the complexity of the EEG [34,35]. In the current study, averaged slope values at high scales tend to increase with the severity of the disease. The opposite tendency was observed at low scales. Previous research aimed to the differentiation between AD patients and healthy controls replicated similar cross-over patterns as temporal scale increases [23,24,44]. For instance, Escudero et al. [24] obtained similar dynamics along all temporal scales comparing AD patients and healthy controls, showing a relation between healthy neurophysiology and low entropy values at high temporal scales. Moreover, Labate et al. [47] studied the relation between multiscale permutation and sample entropy, obtaining similar entropy trends with the scale factor and the progression of the disease. Accordingly with this issue, Mizuno et al. [44] stated that AD may not only be characterized by decreased irregularity in EEG signals, but also by an entropy increase at higher temporal scales. Besides, they indicate the importance of these increments for being pathophysiologically meaningful, because of the correlation with measures of cognitive impairment [44]. This statement agrees with our results, showing a relation between slope values at higher scale calculations and severity of the disease. Changes in long-range neuronal dynamics may be due to disturbances in brain structural pathways, interpreted as abnormal interactions between neuronal systems.

In addition, areas under MSE and rMSSE apparently contain information about cognitive decline. Given two signals, Costa et al. [20] suggested that, if one of them exhibits higher values in most of the time scales, this signal is more complex than the other. However, there is no consensus on the quantitative definition of complexity [20]. Previous research has been aimed to assess EEG complexity by the calculation of areas under MSE curves [34,35]. For this reason, we employed this parameter as an estimator of the term. Therefore, time-series associated with higher areas under MSE and rMSSE curves were considered more complex. Indeed, area values in this study were directly related with the severity of the disease, observing the lowest values on the AD_SEV_ group. Exceptionally, MCI group showed the highest values in both measures when all temporal scales were analyzed. This observation supports that the EEG signals from MCI subjects show the highest degree of resting-state complexity against other subjects. These changes in MCI subjects may be related to an augmented complexity to compensate functional deficits. Evidence of the brain plasticity capabilities attempting to compensate early neurodegeneration in pre-clinical AD patients, even before symptoms begin to emerge, has been previously reported [48]. Jeong [19] remarked complexity decrements in non-linear analyses as evidence that AD patients are less capable of processing information than healthy controls and, hence, as clinical signs of dementia. Therefore, our results can be considered as indicators of neurodegeneration.

### 5.3. Multiscale Entropies for Classification Purposes

The QDA model combining features from both entropy measures achieved higher classification performance than QDA models considering MSE or rMSSE entropy values separately. This point suggests that spectral analysis may provide complementary information to non-linear measures. Regarding with feature selection, a variety of scale factors different than 1 were chosen to build the models. This point implies that multiscale calculations provide data more suitable to categorize classes. Moreover, complexity parameters were also selected, denoting significance in the development of the AD. Previously, other studies were carried out to address AD classification. For instance, Escudero et al. [24] achieved an accuracy of 77.27% discriminating AD patients from control subjects using EEG data from channels P3, P4, O1 and O2 in their MSE study. Fan et al. [23] also calculated their classifier performance using MSE features across 20 temporal scales, obtaining a test accuracy of around 80% with regularized learning methods. Studies aimed at categorizing AD subjects by means of single-scale entropy methods have been also carried out. For instance, Ruiz-Gomez et al. [31] evaluated the discrimination ability of Cross-SampEn from EEG data to categorize controls, MCI subjects and AD patients, obtaining an accuracy of 82.4% in controls vs. AD patients comparison. However, it is important to note that these results come from different databases and, thus, should be interpreted cautiously. AD classification in this study reached a value of 69.7%. An essential aspect of this contribution is the analysis of EEG dynamics among five different classes. This feature is a grade of novelty worth to be considered, since little research has been done assessing AD progression. This is crucial to take into account when evaluating the classification results. It is obvious that the higher is the number of classification groups, the lower is the performance of the classifier. Although our accuracy results are not superior to other studies, its tradeoff for the study of the continuity of the disease is justified. Besides, our classification method showed a 90% chance for a classified AD patient to suffer from MCI or AD. This classification could lead to an AD estimator with high sensibility that may be useful to diagnose the disease.

### 5.4. Limitations

Even though multiscale entropy analyses allow to categorize distinction of neural dynamics of subjects in different AD stages, including its prodromal form, several issues must be taken into account in future studies. Firstly, although we worked on a quite large database, consisting in 253 subjects, significance of the outcomes in a five-group classification could be improved enlarging the sample size. Secondly, a longitudinal study of MCI subjects may be of interest, in order to verify those who progress to AD. This practice would allow gaining new insights on the mechanisms that lead to each type of dementia. Finally, this study is heavily gender unbalanced (177 females against 76 males). This issue might be due to the higher longevity associated with women. Living longer than men and outlasting other death causes, such as infectious diseases, might affect the likelihood to develop dementia eventually, so this phenomenon seems a natural genetic consequence of gender. Additionally, previous work suggested that other physiological gender-dependent causes may lead to higher AD prevalence in women [49].

## 6. Conclusions

This study investigated MSE and rMSSE from EEG signals in MCI and AD patients at 25 temporal scales. Our results suggest that multiscale application of entropy measures can be useful to characterize the underlying effects of neurodegeneration along the AD continuum. Findings provided by this research suggest a high correlation between severity of the AD and parameters of the global analyses in MSE and rMSSE (i.e., averaged slopes and areas under the curves). We can conclude that both entropy analyses are able to identify AD progression. Despite this relationship between MSE and rMSSE, classification performance achieved higher values by combining both measures. This point supports the idea that spectral analyses combined with non-linear analyses may expose underlying components generally hidden. In summary, our results reveal that the disease progression is more evident when it is analyzed from a multiscale approach.

## Figures and Tables

**Figure 1 entropy-21-00544-f001:**
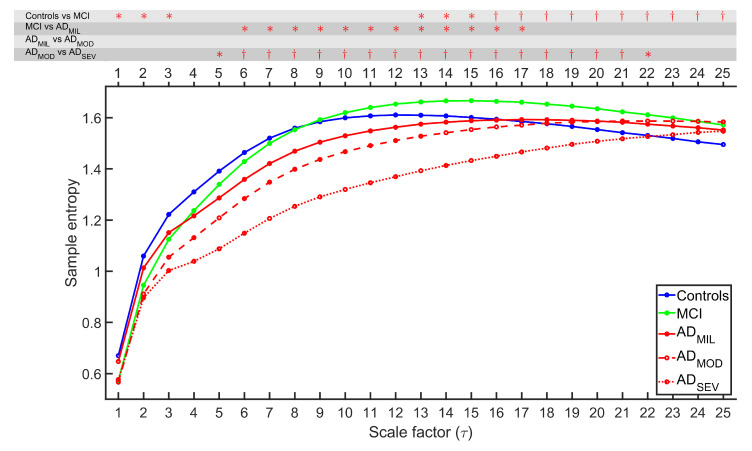
Averaged MSE profiles for each group. On the top of the figure, statistically significant *p*-values are displayed for each scale factor and for each comparison between pairs of consecutive groups (*: *p*-value < 0.05, †: *p*-value < 0.01, FDR-corrected Mann–Whitney *U*-test).

**Figure 2 entropy-21-00544-f002:**
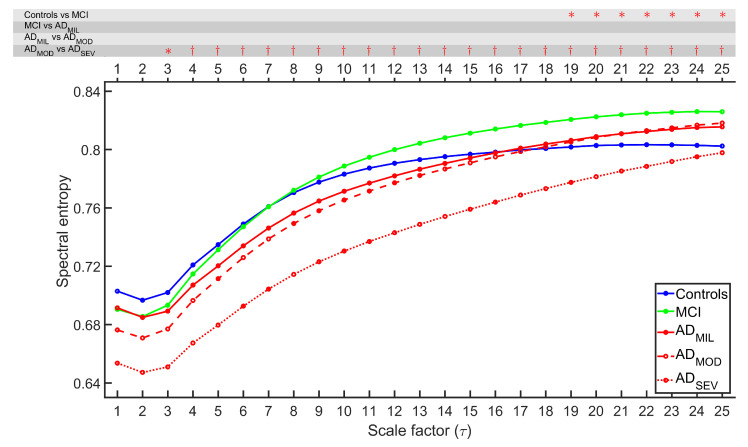
Averaged rMSSE profiles for each group. On the top of the figure, statistically significant *p*-values are displayed for each scale factor and for each comparison between pairs of consecutive groups (*: *p*-value < 0.05, †: *p*-value < 0.01, FDR-corrected Mann–Whitney *U*-test).

**Figure 3 entropy-21-00544-f003:**
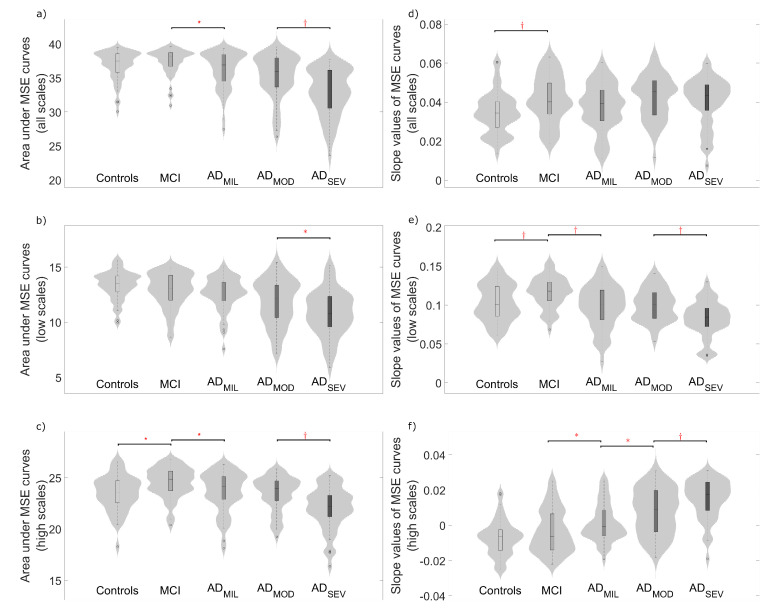
Distribution of complexity parameters: (**a**) area under the whole MSE curve; (**b**) area under the MSE curve for scale factors from 1 to 10; (**c**) area under the MSE curve for scale factors from 11 to 25; (**d**) average slope values of MSE curve; (**e**) average slope values of MSE curve for scale factors from 1 to 10; and (**f**) average slope values of MSE curve for scale factors from 11 to 25. Significant differences along AD continuum are indicated (*: *p*-value < 0.05, †: *p*-value < 0.01, FDR-corrected Mann–Whitney *U*-test).

**Figure 4 entropy-21-00544-f004:**
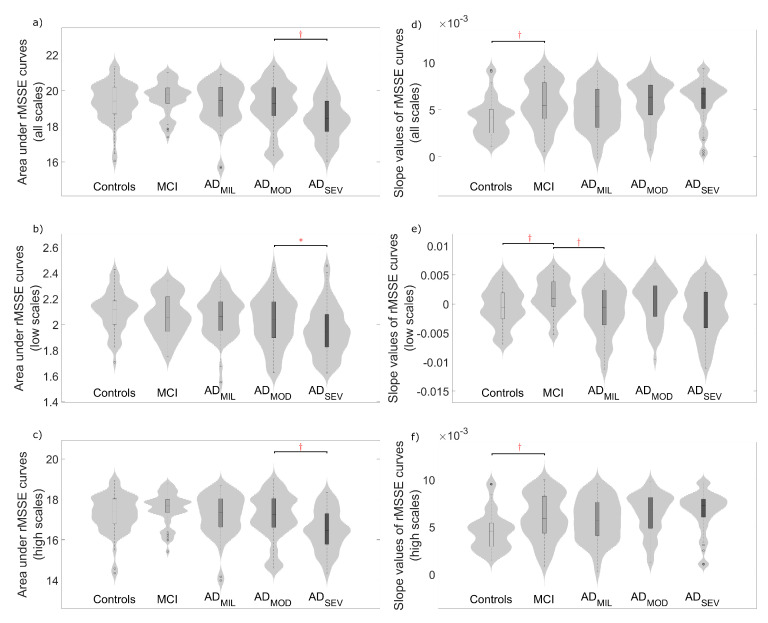
Distribution of complexity parameters: (**a**) area under the whole rMSSE curve; (**b**) area under the rMSSE curve for scale factors from 1 to 3; (**c**) area under the rMSSE curve for scale factors from 4 to 25; (**d**) average slope values of rMSSE curve; (**e**) average slope values of rMSSE curve for scale factors from 1 to 3; and (**f**) average slope values of rMSSE curve for scale factors from 4 to 25. Significant differences along AD continuum are indicated (*: *p*-value < 0.05, †: *p*-value < 0.01, FDR-corrected Mann–Whitney *U*-test).

**Table 1 entropy-21-00544-t001:** Demographic data. SD, standard deviation.

Group	N	Age (Mean ± SD) (Years)	Gender (Female:Male)	MMSE Score (Mean ± SD)
Controls	51	80.14 ± 7.09	25:26	28.82 ± 1.13
MCI subjects	51	85.53 ± 7.25	36:15	23.33 ± 2.84
AD_MIL_ patients	51	80.69 ± 7.05	30:21	22.49 ± 2.27
AD_MOD_ patients	50	81.30 ± 8.04	43:7	13.60 ± 2.76
AD_SEV_ patients	50	79.98 ± 7.82	43:7	2.42 ± 3.70

**Table 2 entropy-21-00544-t002:** Confusion matrix obtained from the QDA classification.

	Est. Class	Controls	MCI Subjects	AD_MIL_ Patients	AD_MOD_ Patients	AD_SEV_ Patients
True Class						
**Controls**		22	16	6	4	3
**MCI subjects**		14	27	2	7	1
**AD_MIL_ patients**		12	14	4	10	11
**AD_MOD_ patients**		4	13	3	15	15
**AD_SEV_ patients**		4	1	1	10	34

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
