# Peer review of "EEG Characterization of the Alzheimer’s Disease Continuum by Means of Multiscale Entropies"

_entropy, 2019, doi:10.3390/e21060544_

Round 1
Reviewer 1 Report
The paper is well written and interesting. It is shown that the measures used together with the multiscale approach reflect differences between the groups considered; so they seem to provide useful tools for investigating Alzheimer’s disease. Background and literature are well investigated and represented. All this justifies publishing the paper in Entropy.
Author Response
We appreciate your effort in reviewing our manuscript. We are grateful for your positive review and we hope you enjoyed reading our paper.
Reviewer 2 Report
In general, the paper is well written, the issue is important, and the database used in the study is very interesting. I only have some suggestions before accepting the paper.
General Comments
- Sample entropy, and thus MSE, is quite sensitive to the sampling rate. For example, if the authors had chosen 1000 Hz instead of 500 Hz, I believe the results could be completely different. Is there any reference, standard or recommendation for using 500 Hz when analyzing EEG signals? Is there any study comparing the effect of different sampling rates on MSE? rMMSE may be less sensitive to it because it does not rely on a pattern-based approach but in the Shannon calculation.
- It seems to make more sense to calculate complexity indices (slope and area under the curve) separated by specific scales, similarly to what Ho and cols did in their paper [PLoS ONE 6(4): e18699]. Notice, for example, that the slope of the curve is dependent on the scale region. From the discussion, it seems that many previous studies have adopted this division.
- Do the authors have any insight into the physiological mechanisms influencing the different time scales?
Specific Comments
- Please, state that Eq. 3 is equivalent to the Shannon entropy of the PSD.
- Lines 228-229: Compared to which group?
- Lines 259-260: The comparison between Control and AD groups is not shown.
- The Discussion section can be shorted, without loss. For example, the paragraph from line 283 to line 291 is part of the fundamentals of the method and not a discussion of findings.
- The first half of the paragraph from lines 292 to 355 is confusing. I cannot differentiate when the authors are talking about previous studies or about their own findings.
- Lines 310-311: As the authors calculated a single slope for the whole MSE or rMMSE curve, they cannot talk about “slope at higher scale”.
Author Response
We appreciate your positive review and comments. The revised manuscript addresses your recommendations, resulting in an improved paper. We have included our responses in the attached document.
